# System Integration Design of High-Performance Piezo-Actuated Fast-Steering Mirror for Laser Beam Steering System

**DOI:** 10.3390/s24216775

**Published:** 2024-10-22

**Authors:** Jung-Gon Kim

**Affiliations:** Ground Technology Research Institute, Agency for Defense Development, Daejeon 34186, Republic of Korea; jgkim78@add.re.kr

**Keywords:** piezoelectric actuator, fast-steering mirror, laser beam steering, tip-tilt control system, adaptive feedforward control, disturbance-attenuation performance

## Abstract

This paper presents an innovative piezo-actuated fast-steering mirror (FSM) that integrates control design and system operation to improve the tracking performance of laser beam steering (LBS) systems. The proposed piezoelectric FSM is centered on two pairs of stacked actuators functioning in the tip-tilt direction via novel flexible hinges with strain-gauge sensors for position measurement. The suggested flexible hinge scheme allows the first fundamental resonance mode with the optical mirror to exceed 400 Hz while achieving an actuation range of ±5 mrad. Thus, the design offers a wider mechanical actuation range than conventional piezoelectric FSMs. Moreover, LBS systems that use fast-steering motion controllers should be robust against dynamic disturbances, such as periodic external vibrations. Such disturbances, inherently associated with the operating conditions for LBS systems, typically reduce the stability of the tip-tilt motion. To attenuate the effects of such disturbances, a high-precision control system is necessary for the tip-tilt motion. Therefore, a control method integrating a proportional–integral controller with an adaptive feedforward control (AFC) algorithm is outlined to enhance tip-tilt tracking performance during high-speed scanning, compared with conventional LBS systems. Based on experimental findings, the AFC algorithm boosted control performance under dynamic disturbances, such as sinusoidal vibrations with multiple frequencies.

## 1. Introduction

Fast-steering mirror (FSM) systems, which enable the tip-tilt motion of a mirror, are widely utilized in high-precision opto-mechatronic devices for optical scanning and pointing applications, such as micro-scanning systems [1], free-space imaging systems [2], adaptive-optics laser systems [3], and beam stabilization systems [4]. Laser beam steering (LBS) systems based on FSMs are increasingly being integrated with moving platforms such as spacecraft, deep-space exploration satellites, airplanes, and other vehicles [5,6,7,8].

FSM systems typically leverage either voice coil [9] or piezoelectric (PZT) actuation [10]. Voice coil actuators support a wide actuation range with moderate bandwidth and can accommodate larger mirror systems [11]. However, the wide actuation range results in a lower actuation resolution compared to piezo-actuated FSMs. Piezo-actuated FSM systems are often relatively compact, using small actuators to support higher system bandwidths and scan rates while operating over narrower scan ranges [12]. Thus, they are suitable for integration into high-speed scanning systems [13]. In addition, they have higher actuation resolutions than voice coil actuators. However, the design elements of piezo-actuated FSM systems include a complex flexible suspension structure that supports the mover and prepresses the actuators [14], mechanical amplification structures for a push–pull configuration [15,16], and specific suspension designs for space-qualified FSMs used on target platforms [17]. Moreover, these systems typically require significant design effort, must be fabricated primarily via electrical discharge machining, and are generally rather challenging to optimize and manufacture [18].

FSM tip-tilt motion systems for LBS must be resilient to dynamic disturbances, such as atmospheric turbulence and mechanical vibrations [19,20]. Atmospheric turbulence, which involves changes in air density between the target and the application platform, is the main cause of degradation in laser propagation and typically has a broadband frequency [21,22,23]. Mechanical vibration is the natural frequency of disturbances (typically narrowband) and can be generated by the laser emission source or by movement of the platform on which the source is mounted [24,25].

Tip-tilt control is designed to accurately aim the laser beam at a target point while suppressing both broadband and narrowband jitter when using an FSM. Fast-convergent adaptive control algorithms for beam jitter suppression have been widely explored to achieve optimal control. Recent studies have introduced least-mean-square (LMS) and recursive least-square adaptive filtering algorithms [26,27,28]. Other jitter-control schemes proposed for PZT actuators include a vibration effect cancellation scheme [29], a notch filter to suppress the dominant resonant mode [30], and a feedforward controller [31]. An input-shaping controller and integral resonance controller have also been developed [32].

In this paper, we aim to design the mechanical structure of the PZT FSM to maximize performance metrics related to mechanical range and main resonant mode frequency (including the optical mirror). Due to the recent need to increase the reliability of piezo actuator FSM motion systems on target platforms under various environmental conditions, the proposed design simplifies the mechanical structure and is relatively easy to manufacture.

Further, a high-precision tip-tilt controller was designed for the FSM system to enable the PZT actuators to maintain an extremely small rotational position under mechanical vibration disturbances [33,34,35]. To this end, it was necessary to design a high-precision FSM motion system that can suppress narrowband jitter using a tip-tilt controller while offering fast response and strong disturbance rejection [36,37,38]. Several methods have been formulated to enhance the dynamic disturbance-attenuation performance of control systems [39,40]. Specifically, iterative control and adaptive feedforward cancellation (AFC) algorithms can cancel out the periodic disturbances produced during operation on the target platform [41,42]. The iterative control algorithm utilizes an internal or external model-based controller design to remove periodic sinusoidal disturbances; this enhances the disturbance-attenuation performance without requiring the feedback controller to be redesigned [43,44]. However, given the hardware-defined narrow stroke range of FSM motion systems, the performance of the classical AFC algorithm is insufficient for precise control of the rotational position.

This paper showcases the design of a high-performance piezo-actuated FSM motion system integrated with a proportional-integral (PI) control system for high-speed steering operation. Based on the results, the mechanical structure of the PZT FSM has been designed and analyzed to maximize performance metrics. Furthermore, designing an effective controller for the piezo-actuated FSM motion system, using a reliable modified AFC algorithm, can reduce control error signals (CES) and enhance vibration compensation for fast tracking. The study demonstrates such an improvement in control performance under multiple sinusoidal vibrations.

The remainder of this paper is organized as follows. Section 2 elaborates on the various system components and design choices for the FSM motion system. Section 3 provides the specifications of the FSM motion system along with its model-based performance analyses. The experiments conducted to validate the proposed control algorithm are described in Section 4. Finally, the conclusions are presented in Section 5.

## 2. Piezo-Actuated FSM System

### 2.1. Overview of Piezo-Actuated FSM System

Figure 1a,b shows an exploded and cross-sectional view of the piezo-actuated FSM without the optical mirror. Figure 2 shows an assembled view of the system without and with the optical mirror. To actuate the mover in both rotational directions, the FSM is equipped with two pairs of stack actuators aligned along the system’s two axes and operated in a push–pull configuration. The optical mirror is directly attachable to the top of the mover, and the mover is suspended from an actuator that provides the prepressing of the piezo stack needed for dynamic behavior. The actuators push against the flexible hinge at the backside of the mover and are mounted on a base plate on the FSM system. The mover is designed to be sufficiently stiff for shifting the structural modes of the moving parts to high frequencies while preventing any deformation of the mirror surface during dynamic operation. This is achieved by attaching a solid ø60 × 80.5 mm aluminum cylinder body to the backside of the mover. However, any rotational motion of the mover in the tip or tilt direction would exert shear and tensile forces on the piezo stacks, which must be avoided as they could cause the stacks to crack. Therefore, the piezo stack actuator is built as a housing assembly to avoid cracking in the shear direction.

A force interface between the rotary mover and each actuator stack, which cannot transmit shear or tensile forces, is established using a flexible hinge bonded to the bottom of the mover and a single contact point at the top of each actuator.

### 2.2. Piezo-Actuated FSM System Design

A flexure is used as the suspension system for the FSM, supporting the mover and the base plate, which are centrally bonded to the front and back sides of the mover. The flexure connects the static and dynamic components of the FSM and is mounted on its body. The flexure also prepresses the piezo-stack actuator assembly to enable dynamic operation. The stroke range of the actuators is expressed as follows:(1)$$∆L\approx L_{0}\left(\frac{k_{A}}{k_{A}+k_{H}}\right),$$
where *k_A_* is the stiffness of the actuator, *k_H_* is the stiffness of the flexure hinge, and *L*_0_ is the range of the unstressed stack actuator. A stiffer flexure design increases the frequency of its structural modes, which is clearly beneficial from a control perspective. Thus, the PZT design has two conflicting aspects that must be considered. First, the PZT stack actuator must be adequately stiff to shift the main resonant mode to higher frequencies. Second, the actuator must be sufficiently compliant to avoid significantly reducing the stroke range.

For the flexure design, a simulation-based parameter study was conducted with various materials and thickness configurations to investigate the tradeoff between range loss and the main resonance mode. To achieve a large system bandwidth, the main resonance mode with the optical mirror was targeted to exceed 400 Hz, while the range loss threshold due to prepressing was set as 5%. With stiffness values of *k_A_* = 12 N/μm and *L*_0_ = 143 μm ± 10% for the maximum range of the stack actuators, a maximum stiffness of 1.48 N/μm was calculated according to Equation (1). As shown in Figure 2, a static mechanical analysis was performed to determine the stiffness *k_H_* of each flexure configuration using ANSYS 2023 R1 Workbench (Ansys Inc., Washington County, PA, USA). A modal analysis revealing the structural modes of each flexure was conducted to attain the main resonance tip mode (equivalent to the tilt mode) of each configuration. The first two modes of the obtained flexure are shown in Figure 3. A main resonance mode (tip/tilt motion) appears at approximately 401 Hz. The overall specifications of piezo-actuated FSM with the optical mirror are shown in Table 1.

### 2.3. Hysteresis of Piezo-Actuated FSM System

For angular control of the piezo-actuated FSM, the mirror angle needs to be measured in the two rotational degrees of freedom. Strain gauge sensors were used to obtain a direct measure of the actual mirror angle based on the strain of the piezo-stack actuators. The mechanical stroke range and hysteresis of the FSM motion system were measured by applying a 1-Hz sinusoidal signal with 15 V amplitude to the piezo-actuated FSM amplifier input of one system axis at a time. The measurement results of both axes (tip and tilt direction) are depicted in Figure 4, demonstrating a mechanical stroke range of approximately ±5 mrad in both tip and tilt direction. The maximum hysteresis is about 15% for both system axes.

## 3. FSM Motion System

### 3.1. Identification of FSM Motion System

In the FSM motion system, the mechanical aspect was comprised of PZT actuators, while the electrical portion consisted of an actuator amplifier and a sensing signal processing unit. As shown in Figure 1, the PZT actuators were designed with a resistance strain gauge sensor attached to the PZT ceramic surface and were mounted below the mirror to enable tip-tilt motion within the required actuation ranges. Figure 5 shows the mirror-mounted FSM using the PZT actuators, encompassing all the mechanical and electrical components required for tip-tilt motion. Designing a feedback control system generally necessitates identifying the dynamic characteristics of the input and output signals of a real plant. The frequency response of the real plant (i.e., the FSM motion system) in this study was analyzed using a dynamic signal analyzer (35670A, KEYSIGHT, Colorado Springs, CO, USA). The identified dynamic specifications of the real plant are summarized in Table 2. Based on these specifications, the frequency responses of the real and modeled plants for the mirror-mounted FSM using PZT actuators are illustrated in Figure 6.

The configuration of the FSM platform for high-precision tip-tilt motion control is illustrated in Figure 7. The FSM motion system was equipped with a mirror-mounted FSM using PZT actuators, a control unit, and an amplifier module. A three-axis vibration generator was used to simulate the moving platform. However, only one axis—the horizontal direction, with gravity as a reference—was considered, as vibrations in this direction have the largest impact on the target platform.

### 3.2. PI Controller

Proportional–integral (PI) controllers are widely used in the operational environment of the target platform for beam jitter control in FSM motion systems. To ensure stability, the tip-tilt controller for FSM motion systems must be designed with the appropriate gain margin, phase margin, 0 dB crossover frequency, and loop gain. The continuous-time and discrete-time transfer functions for a PI controller are denoted by *C_PI_*(*s*) and *C_PI_*(z), respectively, and are expressed as follows:(2)$$C_{PI}\left(s\right)=k_{p}+\frac{k_{i}}{s}=\frac{k_{p}\left(S+\frac{k_{i}}{k_{p}}\right)}{S},$$
(3)$$C_{PI}(z)=k_{p}+\frac{k_{i}T_{s}Z^{-1}}{1-z^{-1}}=\frac{k_{p}-k_{p}Z^{-1}+k_{i}T_{s}Z^{-1}}{1-z^{-1}},$$

Here, *T_s_* is the sampling time, and *k_p_* and *k_i_* respectively represent the control gains of proportional gain and integral gain, respectively. The control gains should be designed considering the robustness and stability of the FSM motion system. In this study, the tunable control gains of the PI controller were experimentally set to *k_p_* = 0.55 and *k_i_* = 1000. Figure 8 depicts the open-loop frequency response of the PI controller designed for the FSM motion system. The gain and phase margins were 11.3 dB and 69.3°, respectively, while the 0 dB crossover frequency and loop gain were 102 Hz and over 30 dB, respectively. The stability gain of the PI controller was selected to achieve the required control performance. The disturbance-attenuation performance of the designed PI controller is presented in Figure 9.

### 3.3. FSM Motion Performance

The closed-loop angle control automatically compensates for the nonlinear characteristics of piezo actuators, such as hysteresis (Figure 4), by comparing the target angle with the actual angle measured by the strain gauge sensor. In this paper, the angle control is based on the optimized PI control loop described in Section 3.2 for the designed pie-zo-actuated FSM.

To verify FSM motion performance, the frequency of the reference signal was set to 1 Hz, and the amplitude at the specified frequency point was varied by a maximum angle of 5 mrad until the current limit or maximum terminal voltage was reached, as depicted in Figure 10. This confirmed the robustness of the PI controller’s performance against hysteresis across the target actuation range. To further verify FSM motion performance, repeated step responses of the FSM motion system were measured, achieving an actuation resolution of 1 µrad root-mean-squared (RMS), as depicted in Figure 11. This indicated that the dynamic step responses are nearly identical across various initial conditions.

## 4. AFC Algorithm for FSM Motion System

During the operation of the AFC algorithm in FSM motion systems, dynamic disturbances are generated by mechanical vibration and atmospheric turbulence. Specifically, disturbances from sudden mechanical vibrations amplify certain frequency components, compromising the control system’s robustness and reducing its convergence speed. Therefore, to enhance the tip-tilt control performance by suppressing these disturbances, an AFC algorithm can be designed to operate in parallel with the feedback controller (i.e., the PI controller), as depicted in Figure 12.

An AFC algorithm is particularly effective in this scenario because it improves the attenuation of periodic disturbances without requiring a redesign of the feedback controller. Theoretically, a disturbance model can be integrated into the feedback loop based on the identified plant model specifications, and this approach can simply be extended to account for multiple sinusoidal components. In addition, the proposed AFC algorithm can be more easily optimized, considering the frequencies of various disturbances.

The disturbance model in the feedback loop of the FSM motion control system is expressed as follows:(4)$$d\left(k\right)=\sum_{i=1}^{n}\left\{a_{i}\left(k\right)\cos\left(\omega kT_{s}\right)+b_{i}\left(k\right)\sin\left(\omega kT_{s}\right)\right\}.$$

*C_AFC_* denotes the AFC algorithm in the feedback loop, while its continuous and discrete forms are denoted by *C_AFC_*(*s*) and *C_AFC_*(*z*), respectively; these forms are expressed as follows:(5)$$\mathrm{Continuous}\;\mathrm{form}:\;C_{AFC}\left(s\right)=g_{i}\left\{\frac{\cos\left(\emptyset_{i}\right)s-\sin\left(\emptyset_{i}\right)\omega_{i}}{s^{2}+\omega_{i}^{2}}\right\},$$
(6)$$\mathrm{Discrete}\;\mathrm{form}\text{:}\;C_{AFC}(z)=g_{i}\left\{\frac{\cos\left(\emptyset_{i}\right)z^{2}-cos(\omega_{i}T_{s}+\emptyset_{i})z}{z^{2}-cos(\omega_{i}T_{s})z+1}\right\},$$
where *T_s_*, *ω_i_*, *g_i_*, and *ϕ_i_* are the sampling time, periodic frequency, adaptive gain, and the phase angle gain of the AFC algorithm, respectively.

For the cancellation of periodic disturbances, the control input of the AFC algorithm adopts the following form in discrete time:(7)$$u_{AFC}\left[k\right]=\hat{d}\left(k\right)=\sum_{i=1}^{n}\left\{\hat{a_{i}}\left[k\right]\cos\left(\omega T_{s}k\right)+\hat{b_{i}}\left[k\right]\sin\left(\omega T_{s}k\right)\right\}.$$

The update laws for the elimination of periodic disturbances are as follows:(8)$$\hat{a_{i}}\left[k\right]=\hat{a_{i}}\left[k-1\right]+g_{i}e\left(k\right)\cos\left(\omega_{i}T_{s}+\emptyset_{i}\right),$$
(9)$$\hat{b_{i}}\left[k\right]=\hat{b_{i}}\left[k-1\right]+g_{i}e\left(k\right)\sin\left(\omega_{i}T_{s}+\emptyset_{i}\right),$$

For sinusoidal compensation, the AFC algorithm can be designed by adjusting *g_i_* and *ϕ_i_*. The adaptation gain *g_i_* requires careful selection through experiments. The disturbance-attenuation performance of the FSM motion system can be improved by increasing the magnitude of the open-loop frequency response at the disturbance frequency of the FSM motion system. However, if *g_i_* is excessively large, the response of the control system at other frequencies will become extremely unstable. Therefore, the fastest convergence (i.e., the fastest elimination of periodic disturbances) must be achieved at a low adaptive gain. Concurrently, it is necessary to select the appropriate *ϕ_i_* at *ω* (periodic disturbance frequency), which is expressed as follows:(10)$$\emptyset_{i}=∠\frac{Ρ(z)}{1+P\left(z\right)C_{PI}(z)}\;at\;\omega_{i},$$
where *P*(z) represents the designed FSM motion system, and *C_PI_*(z) is the PI controller.

The proposed control system design for the FSM motion in this study simultaneously eliminated periodic disturbance and managed vibrations in the experimental environment. Based on the experimental results obtained when utilizing only the PI controller (Figure 13), the periodic disturbances with multiple frequencies in the CES were primarily caused by mechanical vibrations simulating the target platform. The rotational frequency of the platform cooling fan was 59 Hz, and the corresponding frequency of the platform’s mechanical vibration was 73 Hz, which was applied to the FSM motion system with a vibration generator. Therefore, the power spectrum of the CES (Figure 14) shows that the disturbance was caused by the cooling fan’s rotation frequency (i.e., 59 Hz). Another peak was observed at 73 Hz, corresponding to the mechanical vibration of other parts induced by the fan’s rotation.

To eliminate periodic disturbances with multiple frequencies caused by the cooling fan’s rotation and the platform’s mechanical resonance frequencies, an AFC algorithm was incorporated into the FSM motion control system. The resulting frequency responses are presented in Figure 15, Figure 16 and Figure 17.

To validate the proposed control method, the transfer functions of the AFC algorithm for eliminating the aforementioned disturbance frequencies were modeled as follows:(11)$$C_{AFC\_59\mathrm{Hz}}(z)=0.002\times \left\{\frac{-0.34z^{2}+3.37z}{z^{2}-1.99z+1}\right\},$$
(12)$$C_{AFC\_73\mathrm{Hz}}(z)=0.002\times \left\{\frac{-0.77z^{2}+3.37z}{z^{2}-1.99z+1}\right\}..$$

## 5. Experimental Results with AFC for FSM Motion System

The control input was calculated based on the control law in Equation (7). The phase angles of the AFC algorithm at *ω*_1_ = 59 Hz and *ω*_2_ = 73 Hz were estimated to be *ϕ*_1_ = 81° and *ϕ*_2_ = 76°, respectively, based on the update laws in Equations (8) and (9). The experimental results for FSM control using the PI controller, both with and without the AFC algorithm, indicate that the target rotational position was successfully maintained at 0 µrad (i.e., the neutral position). As shown in Figure 6, the residual CES of the PI controller without the AFC algorithm was 12.23 µrad. When the AFC algorithm was used with 59 Hz and 73 Hz disturbances separately, the corresponding root-mean-square (RMS) values of the CES were 7.30 and 9.42 µrad, respectively. These can be observed in Figure 18a,b. When the AFC algorithm was applied with multiple frequency components, i.e., both 59 Hz and 73 Hz concurrently, the RMS of the residual CES was 4.11 µrad, as shown in Figure 18c.

Figure 19a–c demonstrates the power spectra of the AFC algorithm for all the disturbance frequencies. The overall experimental results for the FSM control system with and without the AFC algorithm are compared in Table 3. For fixed-frequency sinusoidal disturbances, the AFC algorithm evidently improved the control effectiveness compared to the PI controller. These results indicate that the AFC algorithm performed efficiently in terms of suppressing periodic disturbances in the FSM system [45,46].

## 6. Conclusions

In this study, the mechanical structure of a PZT FSM was designed and analyzed to maximize the performance metrics linked to the mechanical range and the main resonant mode frequency (including the optical mirror). In addition, the PZT FSM was actuated using two pairs of piezo actuators in a push–pull configuration with a strain gauge sensor system for position measurement. As a result, The PZT FSM allows for an angular range of ±5 mrad in tip and tilt while maintaining a fundamental resonance mode with the optical mirror that exceeds 400 Hz.

Furthermore, an adaptive controller was developed to enable precise beam jitter control for an FSM motion system to suppress the effects of mechanical vibration-induced disturbance on the target platform. The control system incorporated a PI controller and an AFC algorithm operating in parallel to ensure robustness. Experiments revealed that the AFC algorithm improved the disturbance-rejection performance. Specifically, with disturbances of 59 Hz and 73 Hz in the FSM motion system, the CES was reduced by approximately 40% and 23%, respectively. Furthermore, with multiple frequency components, i.e., both 59 Hz and 73 Hz synchronously, the CES was reduced by approximately 66%. Testing in a beam jitter environment that closely resembles the actual target platform revealed that the proposed adaptive control method offers better performance and stability compared to a traditional PI controller without an AFC algorithm. As part of future research, we plan to develop an optical system with a real laser source to assess the performance of the adaptive control algorithm more accurately.

## Figures and Tables

**Figure 1 sensors-24-06775-f001:**
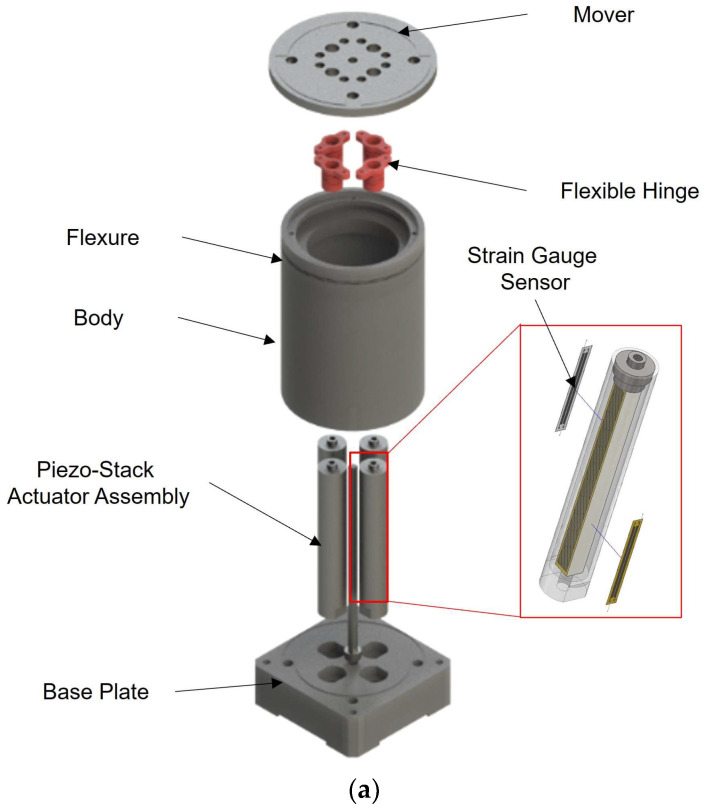
Overview of the piezo-actuated FSM without the optical mirror (which is attached to the mover and operated using two stacked actuators per axis): (**a**) exploded view; (**b**) cross-sectional view.

**Figure 2 sensors-24-06775-f002:**
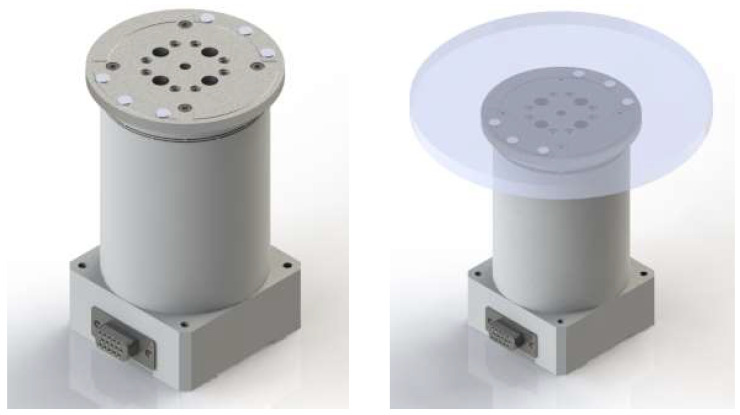
Assembled view of the piezo-actuated FSM without and with the optical mirror.

**Figure 3 sensors-24-06775-f003:**
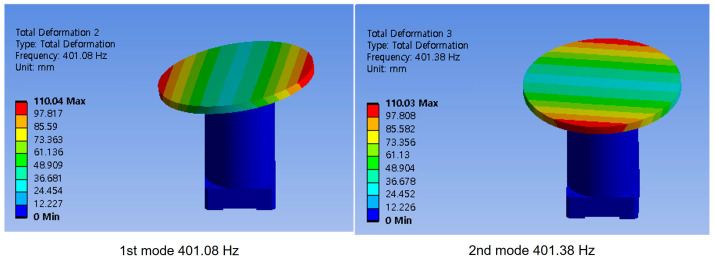
Modal analysis of the piezo-actuated FSM with the optical mirror, showing the first main resonance (tip) mode and the second main resonance (tilt) mode.

**Figure 4 sensors-24-06775-f004:**
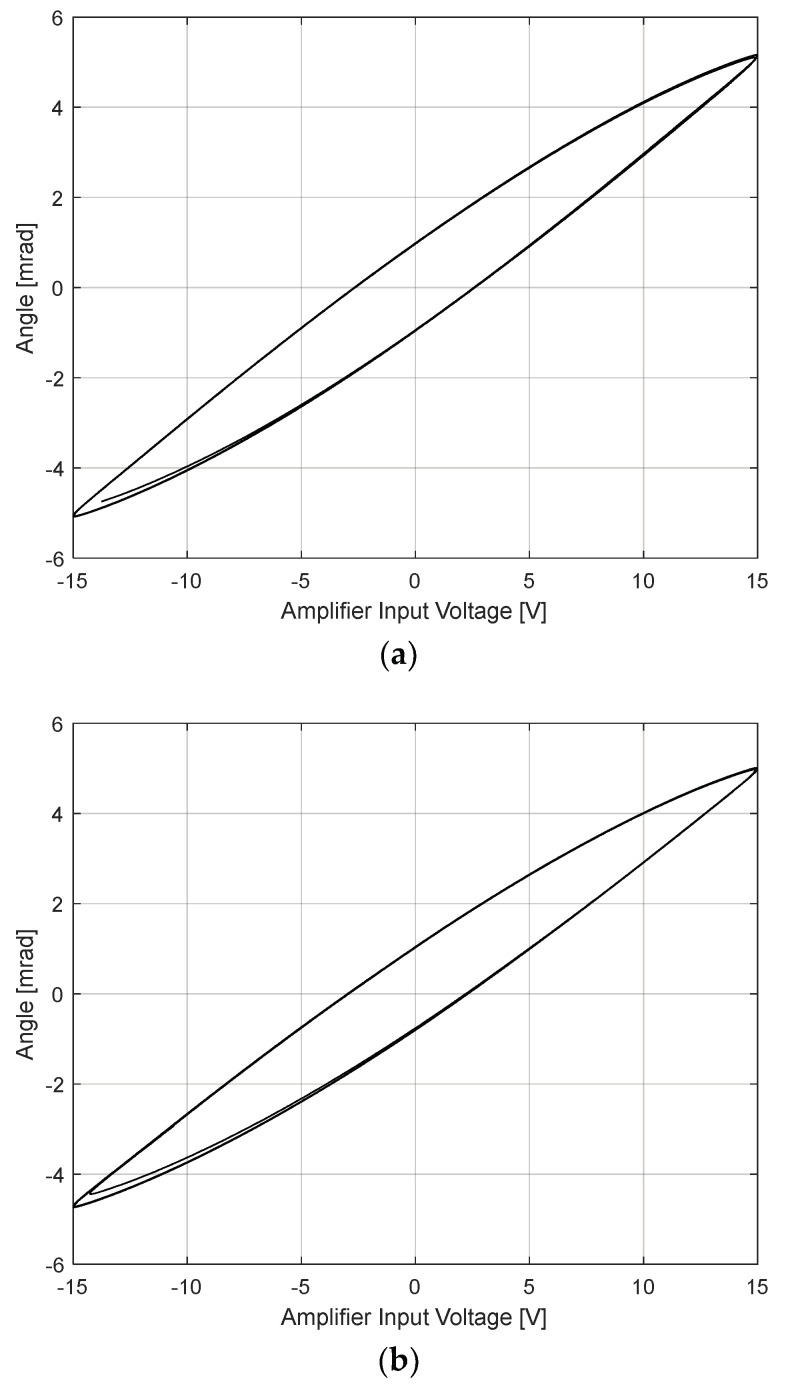
Measured hysteresis for piezo−actuated FSM: (**a**) tip direction; (**b**) tilt direction.

**Figure 5 sensors-24-06775-f005:**
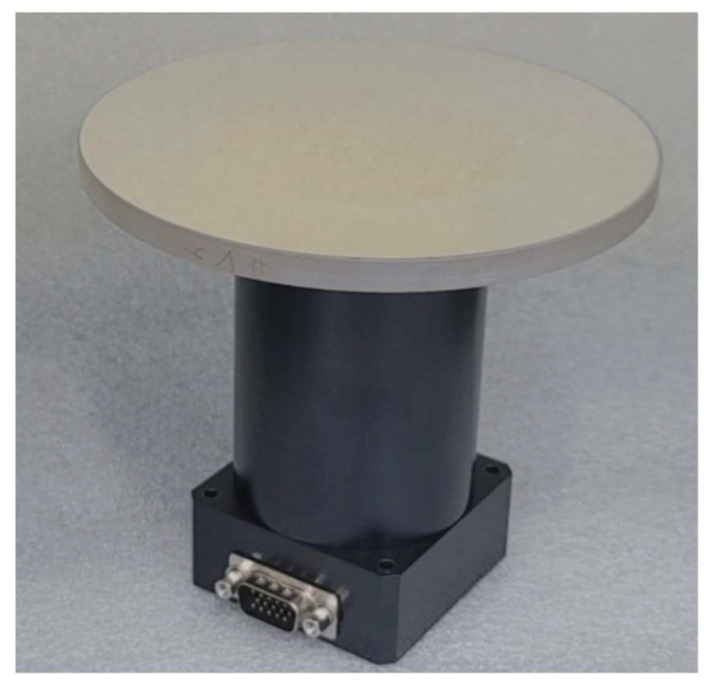
Assembled view of the mirror-mounted FSM using PZT actuator for tip-tilt motion.

**Figure 6 sensors-24-06775-f006:**
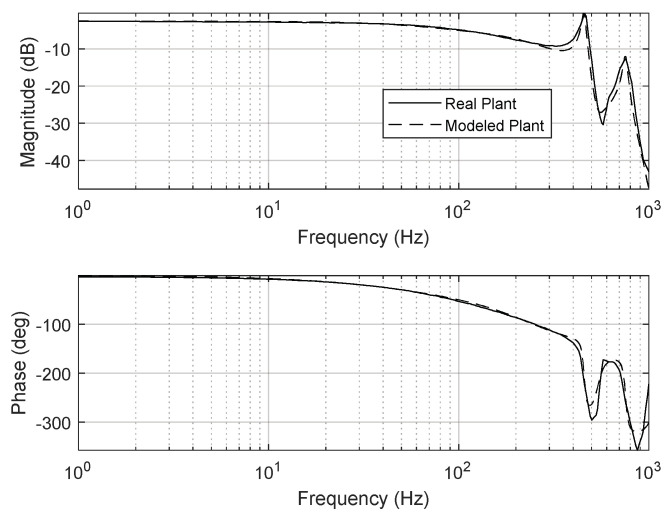
Frequency responses of real and modeled plants for mirror−mounted FSM using PZT actuator.

**Figure 7 sensors-24-06775-f007:**
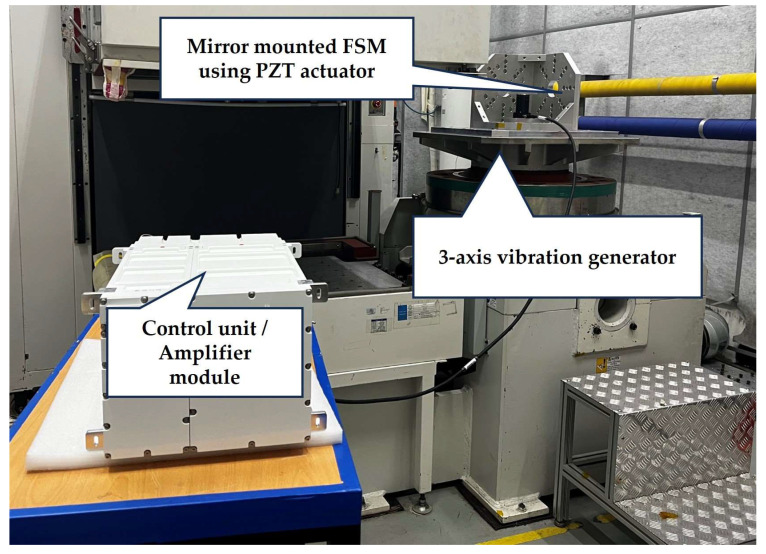
Configuration of FSM platform for high-precision tip-tilt motion control.

**Figure 8 sensors-24-06775-f008:**
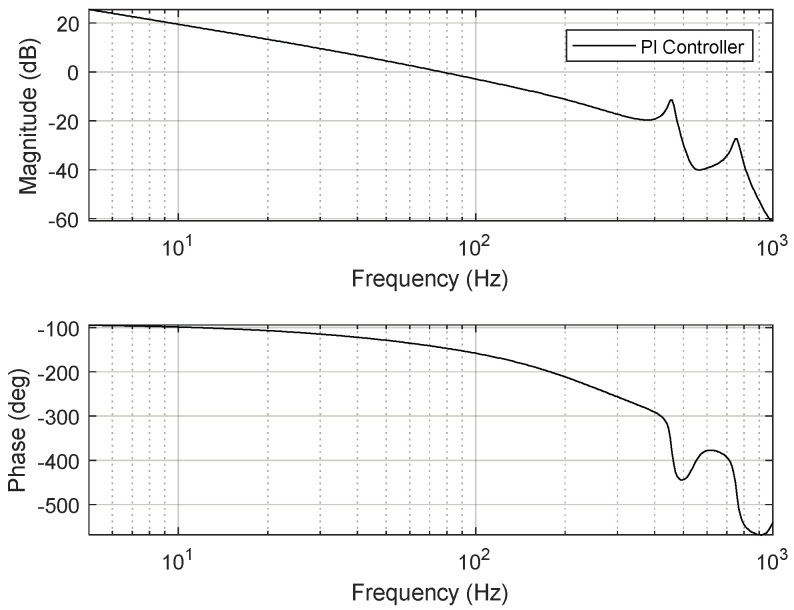
Open−loop frequency response of PI controller for FSM motion system.

**Figure 9 sensors-24-06775-f009:**
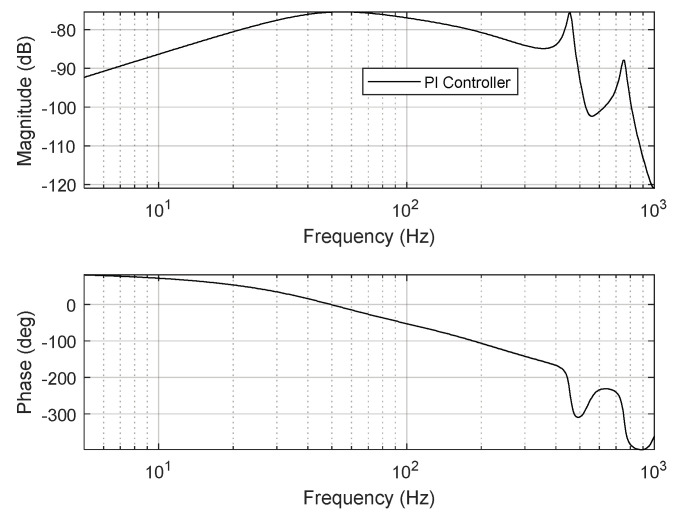
Disturbance−attenuation performance of PI controller for FSM motion system.

**Figure 10 sensors-24-06775-f010:**
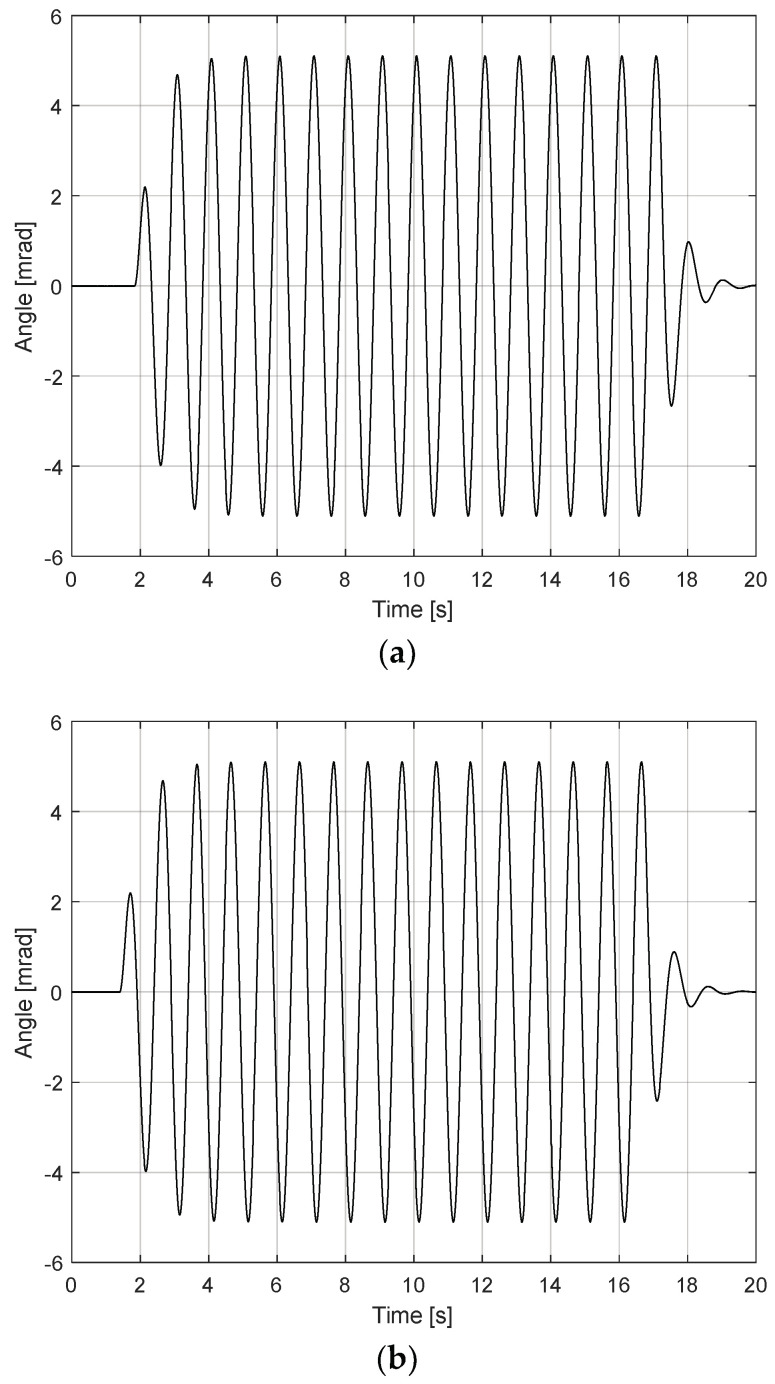
Angular range of FSM motion system: (**a**) tip direction; (**b**) tilt direction.

**Figure 11 sensors-24-06775-f011:**
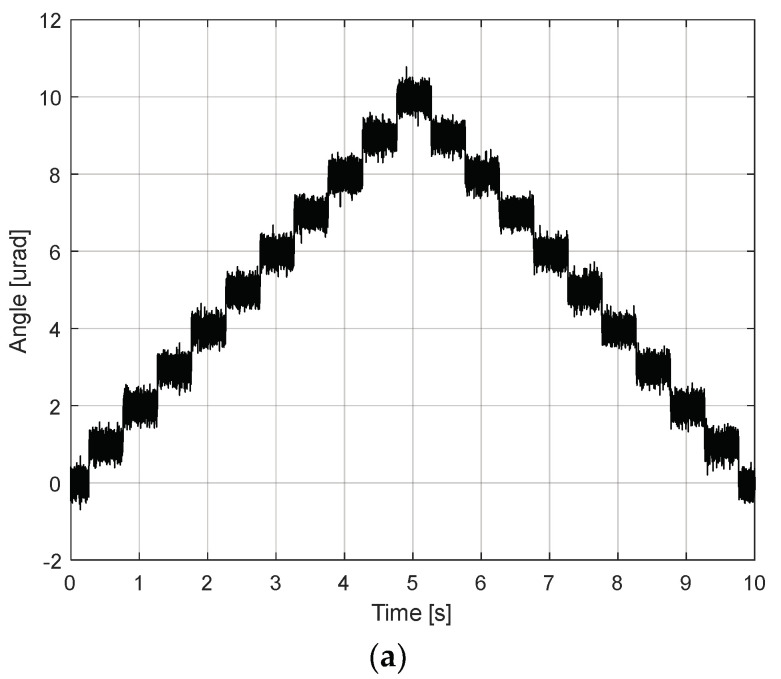
Actuating resolution of FSM motion system: (**a**) tip direction; (**b**) tilt direction.

**Figure 12 sensors-24-06775-f012:**
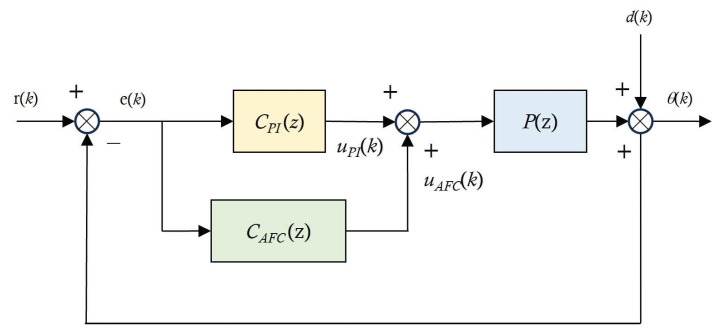
Schematic of FSM control system with AFC algorithm.

**Figure 13 sensors-24-06775-f013:**
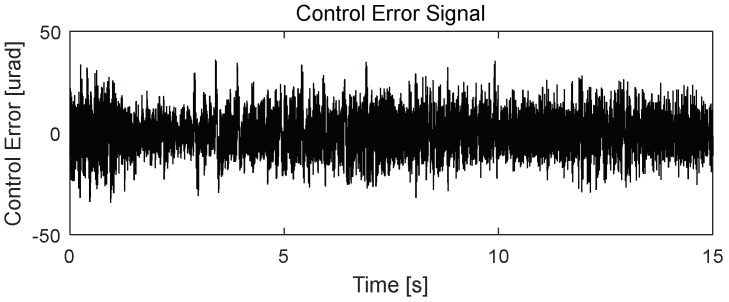
Performance of PI controller for FSM motion system (residual CES).

**Figure 14 sensors-24-06775-f014:**
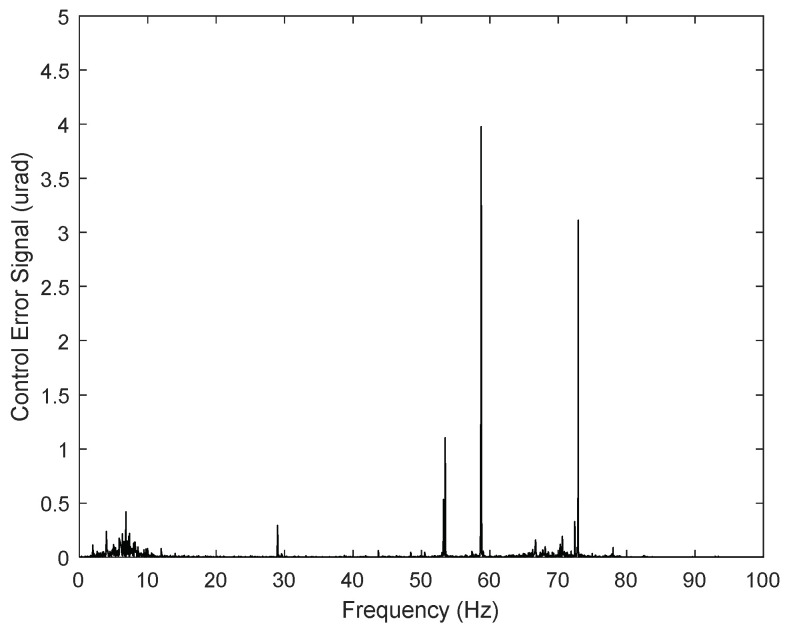
Performance of PI controller for FSM motion system (power spectrum).

**Figure 15 sensors-24-06775-f015:**
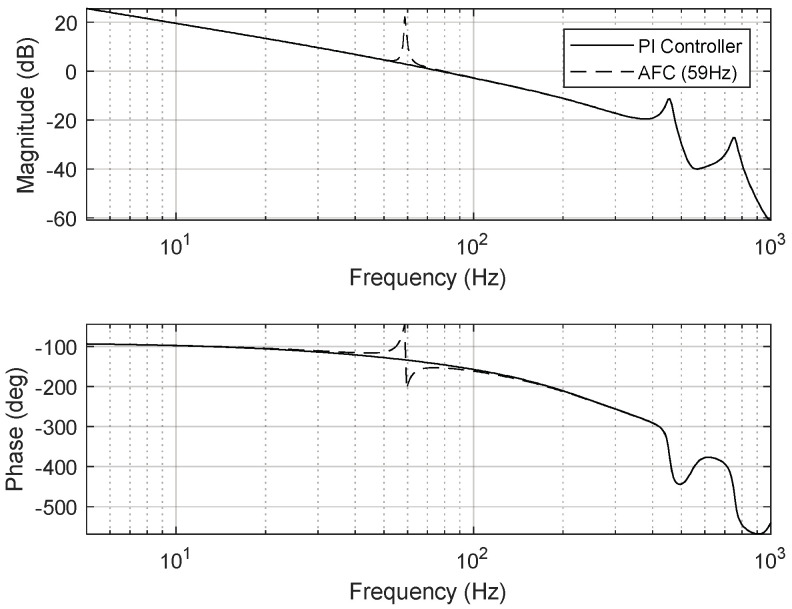
Open−loop frequency response of FSM motion system (AFC at 59 Hz).

**Figure 16 sensors-24-06775-f016:**
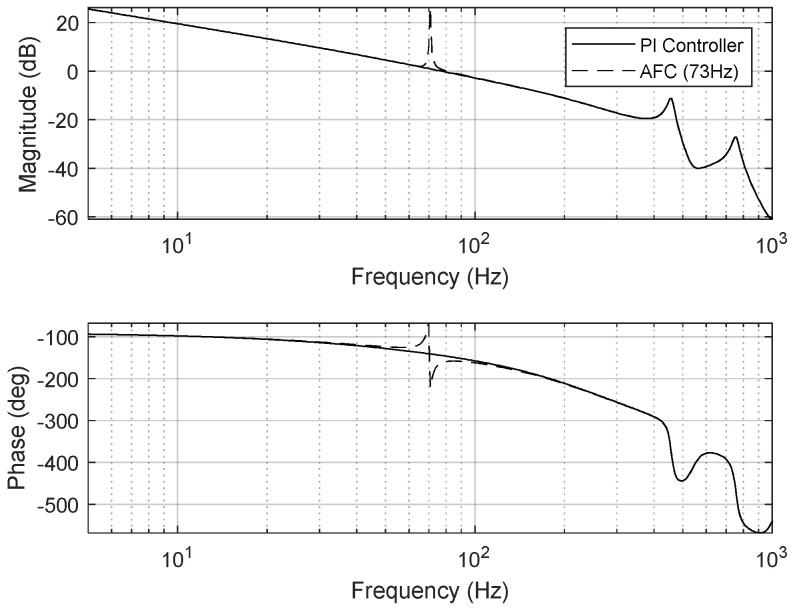
Open−loop frequency response of FSM motion system (AFC at 73 Hz).

**Figure 17 sensors-24-06775-f017:**
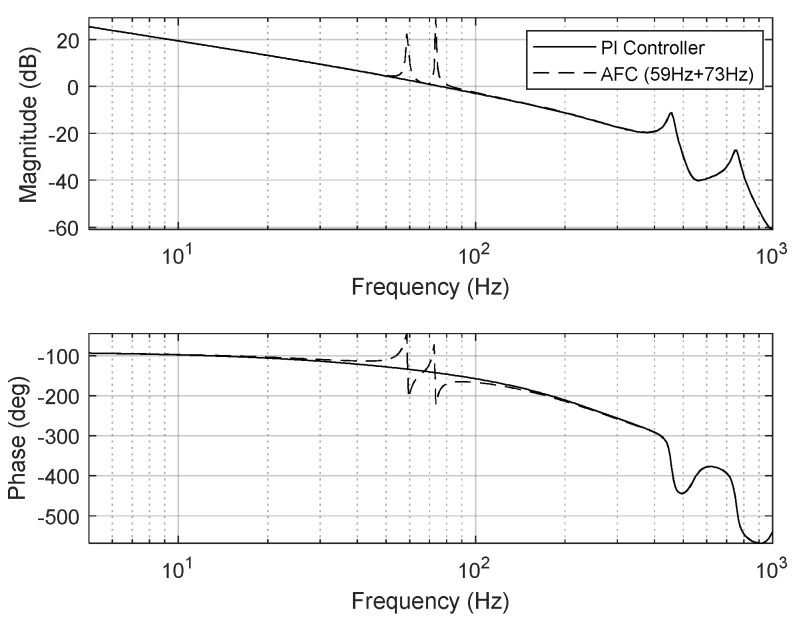
Open−loop frequency response of FSM motion system (AFC at 59 Hz and 73 Hz).

**Figure 18 sensors-24-06775-f018:**
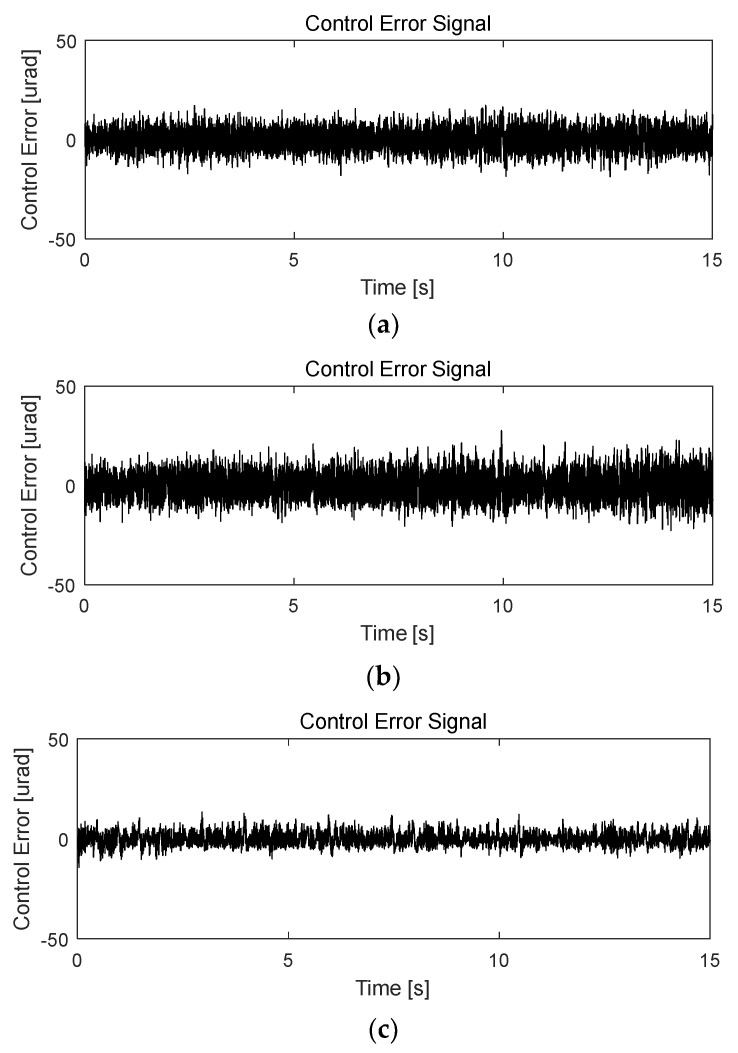
Performance of AFC algorithm for FSM motion system (residual CES): (**a**) 59 Hz; (**b**) 73 Hz; (**c**) 59 Hz and 73 Hz.

**Figure 19 sensors-24-06775-f019:**
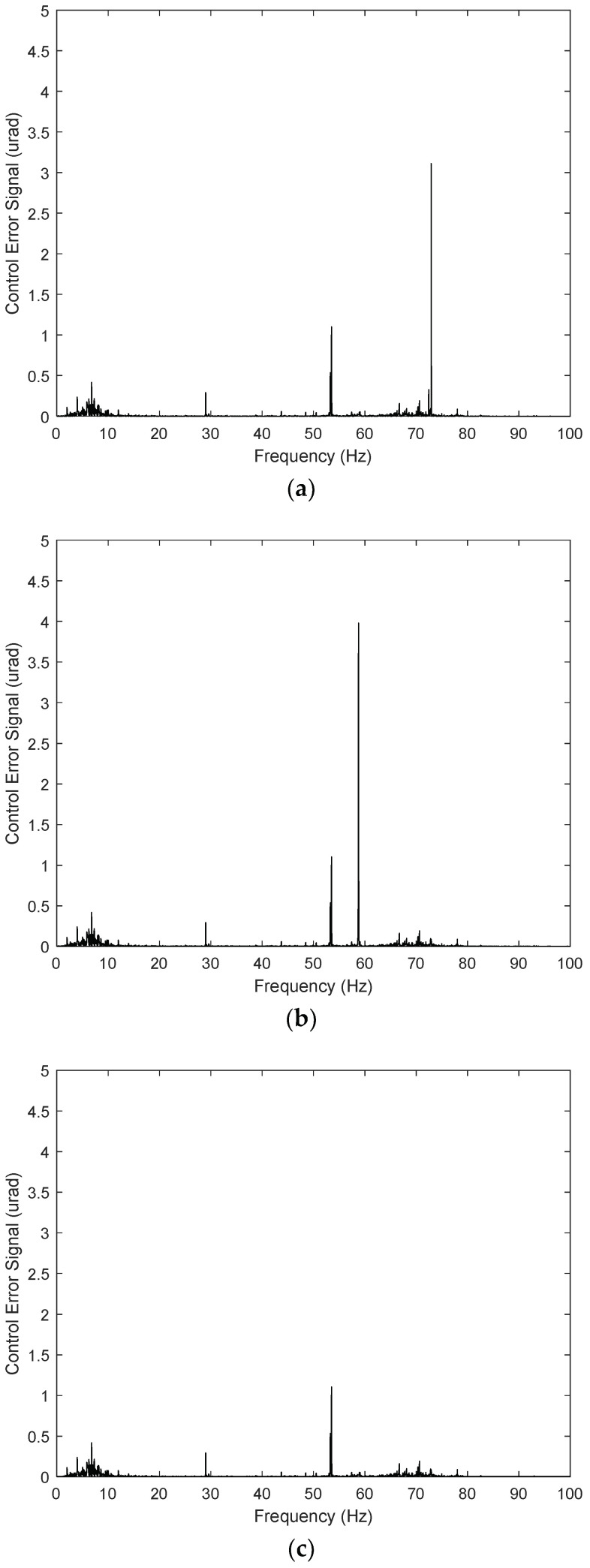
Performance of AFC algorithm for FSM motion system (power spectrum): (**a**) 59 Hz; (**b**) 73 Hz; (**c**) 59 Hz and 73 Hz.

**Table 1 sensors-24-06775-t001:** Dynamic specifications of the real plant.

Identified Specification	Value
Stroke range	±5 mrad
Angular resolution	1 µrad
Mover diameter	70 mm
Resonance frequency (tip)	401.08 Hz
Resonance frequency (tilt)	401.38 Hz

**Table 2 sensors-24-06775-t002:** Dynamic specifications of piezo-actuated FSM.

Identified Specification	Value
Resonance frequency	407.8 Hz
5 Hz sensitivity	178.9 µrad/V
Voltage amplifier gain	15 V/V
Sensor amplifier gain	0.006 V/µrad

**Table 3 sensors-24-06775-t003:** Overall control performance for FSM motion system with and without AFC algorithm.

Specification	PI Controller	AFC (59 Hz)	AFC (73 Hz)	AFC (59 & 73 Hz)
Control error (RMS)	12.23 μrad	7.30 μrad	9.42 μrad	4.11 μrad
Error reduction	–	40.31%	22.98%	66.39%

## Data Availability

Data is contained within the article.

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
