# Peer review of "System Integration Design of High-Performance Piezo-Actuated Fast-Steering Mirror for Laser Beam Steering System"

_sensors, 2024, doi:10.3390/s24216775_

Round 1

Reviewer 1 Report

Comments and Suggestions for Authors

This manuscript presents a system integration design of piezo-actuated fast-steering mirror. Authors describe the detailed working principle and conduct the finite element analysis. Experimental results validate the proposed control method integrating a proportionalintegral controller with an adaptive feedforward control (AFC) algorithm can work well. However, several concerns must be addressed:

1. What is the novelty in structure and working principle of the system integration design of piezo-actuated fast-steering mirror? As far as I am concerned, the structure and working principle are the same with the traditional piezo-actuated FSM.

2. Compared to commercial products such as PI, what are the advantages of the piezo-actuated FSM in this work?

3. The system integration design is a contribution of this work; however, the calibration of the strain gauge sensor is not stated. Moreover, is the annotation of the strain gauge sensor incorrect in Fig. 1?

4. What is the design objective of the piezo-actuated FSM? and it needs to be clearly defined.

5. In the experimental Section, the angular range and resolution tests are missed, which are important characteristics to verify the design.

Author Response

Comments 1: What is the novelty in structure and working principle of the system integration design of piezo-actuated fast-steering mirror? As far as I am concerned, the structure and working principle are the same with the traditional piezo-actuated FSM.

Response 1: First things first. Thank you for your review comments. The proposed design is an optimal design with enhanced mechanical stiffness to ensure reliability under the influence of environmental characteristics for the target platform application. Although the overall mechanical structure design concept may appear similar, it is not the geometry of the tip-tilt flexible hinge connected to the four piezoelectric actuators, it is the piezoelectric FSM motion system with mirror that meets the required performance (actuation range, actuation resolution, etc.) of the target platform. 

Comments 2: Compared to commercial products such as PI, what are the advantages of the piezo-actuated FSM in this work?

Response 2: As answered in review comment #1, although the overall mechanical structure design concept may appear similar, the mechanical rigidity is enhanced to ensure reliability under the influence of environmental characteristics for the target platform application by using a tip-tilt flexible hinge connected with four piezoelectric actuators.

Comments 3: The system integration design is a contribution of this work; however, the calibration of the strain gauge sensor is not stated. Moreover, is the annotation of the strain gauge sensor incorrect in Fig. 1?

Response 3: I agree with your review comment. First of all, there was an error in the annotation of the strain gauge sensor mentioned in Figure 1, which has been corrected. Moreover, added the results of the calibration of the strain gauge sensor to section 2.3 and 3.3.

Comments 4: What is the design objective of the piezo-actuated FSM? and it needs to be clearly defined.

Response 4: I agree with your review comment. Therefore, revised the design objective of the piezo-actuated FSM in the introduction. 

Comments 5: In the experimental Section, the angular range and resolution tests are missed, which are important characteristics to verify the design.

Response 5:  I agree with your review and have therefore added the drive range and drive resolution results to Section 3.3.

Reviewer 2 Report

Comments and Suggestions for Authors

This paper is mainly about the design, analysis and control of a piezoelectric fast-steering mirror for laser beam steering system. The topic is interesting, and the piezoelectric fast-steering mirror is a good topic as it has some important application in system of laser beam control. However, after carefully reading the whole paper, I cannot give a recommendation for publication as the novelty and contribution of this work is very limited.

Firstly, it seems that the author just finished a very routine design and experiment work as the piezoelectric fast-steering mirror discussed in this paper is just the same with the previous design, and there is nearly no innovations for the design of the proposed design.

Secondly, for the performances of the designed piezoelectric fast-steering mirror, it is really hard to say that they are better than the products of some companies. The stroke range of about ±5 mrad is not large, the basic frequency of about 400 Hz is not high, furthermore, the control error of about 4 μrad is also large.

Thirdly, the literature review of this paper is somewhat out of date, it seems that the authors have not an in-depth investigation about the recent progress about the piezoelectric fast-steering mirror.

Comments on the Quality of English Language

no comments

Author Response

Comments 1: it seems that the author just finished a very routine design and experiment work as the piezoelectric fast-steering mirror discussed in this paper is just the same with the previous design, and there is nearly no innovations for the design of the proposed design.

Response 1: Thank you for your review comments. I partially agree with your comment that the proposed design is not innovative, because as you know, the motion system using piezoelectric high-speed steering mirrors should be designed according to the required characteristics of the target platform.

the proposed piezoelectric FSM design is an optimal design with increased mechanical stiffness to ensure reliability under the influence of environmental characteristics for the target platform application. Even though the overall mechanical structure design concept looks similar, it is not the shape of a tip-tilt flexible hinge connected to four piezoelectric actuators, but a piezoelectric FSM motion system with a mirror that meets the required performance (actuation range, actuation resolution, etc.) of the target platform.

Comments 2: for the performances of the designed piezoelectric fast-steering mirror, it is really hard to say that they are better than the products of some companies. The stroke range of about ±5 mrad is not large, the basic frequency of about 400 Hz is not high, furthermore, the control error of about 4 μrad is also large.

Response 2:  I partially agree with your review comment that the control error of 4 μrad is large for a stroke range of ±5 mrad. However, as you know, the control error of FSM motion systems generally varies with the stroke range. If the test measurement environment is made on an optical table and the stroke range is within ±1 mrad, the control error will be much smaller than 4 μrad. One of the reasons for this is that the angular output signal processing is easier with a smaller stroke range. For the purposes of this paper, I do not believe that the control error of an FSM motion system with a stroke range of ±5 mrad under vibration applied by a 3-axis vibration generator, which is the actual dynamic condition of the target platform, is larger than about 4 μrad. In future work, I will design a system that will allow the control error of the system to be smaller than about 4 μrad.

Comments 3: the literature review of this paper is somewhat out of date, it seems that the authors have not an in-depth investigation about the recent progress about the piezoelectric fast-steering mirror.

Response 3: 
I agree with your review comments. Therefore, I have updated the references in the paper to reflect
recent research trends. (reference numbers 8, 11, 12, 14, 22, 23, 24, 29, 40, 41, 42, 43)

Reviewer 3 Report

Comments and Suggestions for Authors

Main Contribution

  • The paper presents a novel piezo-actuated fast-steering mirror (FSM) designed to enhance the tracking performance of laser beam steering (LBS) systems. It integrates control design with system operation to improve performance under dynamic disturbances, such as periodic vibrations.
  • The proposed FSM utilizes two pairs of stacked actuators and strain-gauge sensors, achieving a mechanical drive range of ±5 mrad and a fundamental resonance mode exceeding 400 Hz, which is superior to conventional designs.

Suggestions for Improvement

  1. Include more diverse experimental conditions to validate the control algorithms under various operational scenarios.
  2. Provide a comprehensive analysis of performance metrics, such as response time and accuracy, to better illustrate improvements.
  3. Include comparisons with existing FSM designs to highlight advantages and limitations more clearly.
  4. Enhance figures and diagrams for better clarity, especially in illustrating the mechanical structure and control system.
  5. Discuss potential optimizations for the adaptive feedforward control (AFC) algorithm to improve its efficiency.
  6. Conduct long-term stability tests to assess the durability and reliability of the proposed system.
  7. Include a cost analysis of the proposed design versus traditional systems to evaluate economic feasibility.
  8. Explore potential applications beyond LBS systems to attract a wider audience.
  9. Consider integrating AI techniques for predictive control to enhance performance further.
  10. Propose a feedback mechanism for users to report performance issues, which could guide future improvements.

Author Response

Comments 1: Include more diverse experimental conditions to validate the control algorithms under various operational scenarios.

Response 1: In this paper, the results of a study on the application of the proposed FSM motion system and control algorithm under dynamic vibration applied to a target platform are presented.

Comments 2: Provide a comprehensive analysis of performance metrics, such as response time and accuracy, to better illustrate improvements.

Response 2:  In Section 3.3, I further mentioned the angular range and actuating resolution as control performance.

Comments 3: Include comparisons with existing FSM designs to highlight advantages and limitations more clearly.

Response 3:  the mechanical rigidity of piezo-actuated FSM motion system is enhanced to ensure reliability under the influence of environmental characteristics for the target platform application by using a tip-tilt flexible hinge connected with four piezoelectric actuators.

Comments 4: Enhance figures and diagrams for better clarity, especially in illustrating the mechanical structure and control system.

Response 4:  I agree with your review. Therefore, I have added a cross-sectional view in Figure 1.

Comments 5: Discuss potential optimizations for the adaptive feedforward control (AFC) algorithm to improve its efficiency.

Response 5:  Potential optimizations to improve the efficiency of the adaptive feedforward control (AFC) algorithm, with further research to propose improved control algorithms.

Comments 6: Conduct long-term stability tests to assess the durability and reliability of the proposed system.

Response 6:  To ensure reliability for the target platform application, the mechanical stiffness of the piezoelectrically actuated FSM motion system was improved, and experimental results were obtained under the influence of dynamic disturbances.

Comments 7: Include a cost analysis of the proposed design versus traditional systems to evaluate economic feasibility.

Response 7:  I agree with your review that it would be more fulfilling to include a cost analysis, but I hope you understand that this is not possible.

Comments 8: Explore potential applications beyond LBS systems to attract a wider audience.

Response 8:  An improved design is currently underway for application in multi-axis laser beam steering (LBS) systems.

Comments 9: Consider integrating AI techniques for predictive control to enhance performance further.

Response 9:  I completely agree with your review comments. However, integrating AI technology is not easy at the moment, but we will consider it for future research.

Comments 10: Propose a feedback mechanism for users to report performance issues, which could guide future improvements.

Response 10:  In the future, we plan to further configure the optical system using lasers and conduct research on topics related to improved jitter control.

Round 2

Reviewer 1 Report

Comments and Suggestions for Authors

The authors have answered my concerns and revised this paper. It can be accepted for publication.

Reviewer 3 Report

Comments and Suggestions for Authors

The authors considered my comments and suggestions. Good luck.